# Towards Biologically Plausible Convolutional Networks

**Roman Pogodin**
Gatsby Unit, UCL
`roman.pogodin.17@ucl.ac.uk`

**Yash Mehta**
Gatsby Unit, UCL
`y.mehta@ucl.ac.uk`

**Timothy P. Lillicrap**
DeepMind; CoMPLEX, UCL
`countzero@google.com`

**Peter E. Latham**
Gatsby Unit, UCL
`pel@gatsby.ucl.ac.uk`

## Abstract

Convolutional networks are ubiquitous in deep learning. They are particularly useful for images, as they reduce the number of parameters, reduce training time, and increase accuracy. However, as a model of the brain they are seriously problematic, since they require weight sharing – something real neurons simply cannot do. Consequently, while neurons in the brain can be locally connected (one of the features of convolutional networks), they cannot be convolutional. Locally connected but non-convolutional networks, however, significantly underperform convolutional ones. This is troublesome for studies that use convolutional networks to explain activity in the visual system. Here we study plausible alternatives to weight sharing that aim at the same regularization principle, which is to make each neuron within a pool react similarly to identical inputs. The most natural way to do that is by showing the network multiple translations of the same image, akin to saccades in animal vision. However, this approach requires many translations, and doesn't remove the performance gap. We propose instead to add lateral connectivity to a locally connected network, and allow learning via Hebbian plasticity. This requires the network to pause occasionally for a sleep-like phase of "weight sharing". This method enables locally connected networks to achieve nearly convolutional performance on ImageNet and improves their fit to the ventral stream data, thus supporting convolutional networks as a model of the visual stream.

## 1 Introduction

Convolutional networks are a cornerstone of modern deep learning: they're widely used in the visual domain [1, 2, 3], speech recognition [4], text classification [5], and time series classification [6]. They have also played an important role in enhancing our understanding of the visual stream [7]. Indeed, simple and complex cells in the visual cortex [8] inspired convolutional and pooling layers in deep networks [9] (with simple cells implemented with convolution and complex ones with pooling). Moreover, the representations found in convolutional networks are similar to those in the visual stream [10, 11, 12, 13] (see [7] for an in-depth review).

Despite the success of convolutional networks at reproducing activity in the visual system, as a model of the visual system they are somewhat problematic. That's because convolutional networks share weights, something biological networks, for which weight updates must be local, can't do [14]. Locally connected networks avoid this problem by using the same receptive fields as convolutional networks (thus locally connected), but without weight sharing [15]. However, they pay a price for biological plausibility: locally connected networks are known to perform worse than their

35th Conference on Neural Information Processing Systems (NeurIPS 2021).

convolutional counterparts on hard image classification tasks [15, 16]. There is, therefore, a need for a mechanism to bridge the gap between biologically plausible locally connected networks and implausible convolutional ones.

Here, we consider two such mechanisms. One is to use extensive data augmentation (primarily image translations); the other is to introduce an auxiliary objective that allows some form of weight sharing, which is implemented by lateral connections; we call this approach dynamic weight sharing.

The first approach, data augmentation, is simple, but we show that it suffers from two problems: it requires far more training data than is normally used, and even then it fails to close the performance gap between convolutional and locally connected networks. The second approach, dynamic weight sharing, implements a sleep-like phase in which neural dynamics facilitate weight sharing. This is done through lateral connections in each layer, which allows subgroups of neurons to share their activity. Through this lateral connectivity, each subgroup can first equalize its weights via anti-Hebbian learning, and then generate an input pattern for the next layer that helps it to do the same thing. Dynamic weight sharing doesn't achieve perfectly convolutional connectivity, because in each channel only subgroups of neurons share weights. However, it implements a similar inductive bias, and, as we show in experiments, it performs almost as well as convolutional networks, and also achieves better fit to the ventral stream data, as measured by the Brain-Score [12, 17].

Our study suggests that convolutional networks may be biologically plausible, as they can be approximated in realistic networks simply by adding lateral connectivity and Hebbian learning. As convolutional networks and locally connected networks with dynamic weight sharing have similar performance, convolutional networks remain a good "model organism" for neuroscience. This is important, as they consume much less memory than locally connected networks, and run much faster.

## 2 Related work

Studying systems neuroscience through the lens of deep learning is an active area of research, especially when it comes to the visual system [18]. As mentioned above, convolutional networks in particular have been extensively studied as a model of the visual stream (and also inspired by it) [7], and also as mentioned above, because they require weight sharing they lack biological plausibility. They have also been widely used to evaluate the performance of different biologically plausible learning rules [15, 19, 20, 21, 22, 23, 24, 25].

Several studies have tried to relax weight sharing in convolutions by introducing locally connected networks [15, 16, 26] ([26] also shows that local connectivity itself can be learned from a fully connected network with proper weight regularization). Locally connected networks perform as well as convolutional ones in shallow architectures [15, 26]. However, they perform worse for large networks and hard tasks, unless they're initialized from an already well-performing convolutional solution [16] or have some degree of weight sharing [27]. In this study, we seek biologically plausible regularizations of locally connected network to improve their performance.

Convolutional networks are not the only deep learning architecture for vision: visual transformers (e.g., [28, 29, 30, 31]), and more recently, the transformer-like architectures without self-attention [32, 33, 34], have shown competitive results. However, they still need weight sharing: at each block the input image is reshaped into patches, and then the same weight is used for all patches. Our Hebbian-based approach to weight sharing fits this computation as well (see Appendix A.4).

## 3 Regularization in locally connected networks

### 3.1 Convolutional versus locally connected networks

Convolutional networks are implemented by letting the weights depend on the difference in indices. Consider, for simplicity, one dimensional convolutions and a linear network. Letting the input and output of a one layer in a network be $x_j$ and $z_i$, respectively, the activity in a convolutional network is

$$z_i = \sum_{j=1}^{N} w_{i-j} x_j \, , \tag{1}$$

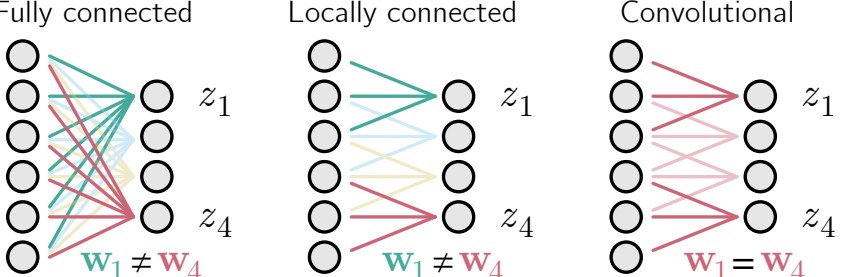

Figure 1: Comparison between layer architectures: fully connected (left), locally connected (middle) and convolutional (right). Locally connected layers have different weights for each neuron $z_1$ to $z_4$ (indicated by different colors), but have the same connectivity as convolutional layers.

where $N$ is the number of neurons; for definiteness, we'll assume $N$ is the same in in each layer (right panel in Fig. 1). Although the index $j$ ranges over all $N$ neurons, many, if not most, of the weights are zero: $w_{i-j}$ is nonzero only when $|i-j| \leq k/2 < N$ for kernel size $k$.

For networks that aren't convolutional, the weight matrix $w_{i-j}$ is replaced by $w_{ij}$,

$$z_i = \sum_{j=1}^{N} w_{ij} x_j \,. \tag{2}$$

Again, the index $j$ ranges over all $N$ neurons. If all the weights are nonzero, the network is fully connected (left panel in Fig. 1). But, as in convolutional networks, we can restrict the connectivity range by letting $w_{ij}$ be nonzero only when $|i-j| \leq k/2 < N$, resulting in a *locally connected*, but non-convolutional, network (center panel in Fig. 1).

### 3.2 Developing convolutional weights: data augmentation versus dynamic weight sharing

Here we explore the question: is it possible for a locally connected network to develop approximately convolutional weights? That is, after training, is it possible to have $w_{ij} \approx w_{i-j}$? There is one straightforward way to do this: augment the data to provide multiple translations of the same image, so that each neuron within a channel learns to react similarly (Fig. 2A). A potential problem is that a large number of translations will be needed. This makes training costly (see Section 5), and is unlikely to be consistent with animal learning, as animals see only a handful of translations of any one image.

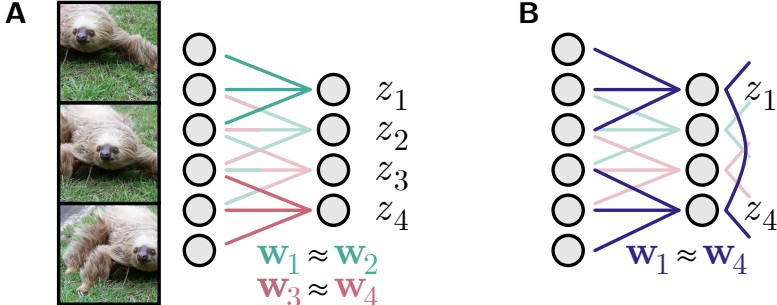

Figure 2: Two regularization strategies for locally connected networks. **A.** Data augmentation, where multiple translations of the same image are presented simultaneously. **B.** Dynamic weight sharing, where a subset of neurons equalizes their weights through lateral connections and learning.

A less obvious solution is to modify the network so that during learning the weights become approximately convolutional. As we show in the next section, this can be done by adding lateral connections, and introducing a *sleep phase* during training (Fig. 2B). This solution doesn't need more data, but it does need an additional training step.

# 4 A Hebbian solution to dynamic weight sharing

If we were to train a locally connected network without any weight sharing or data augmentation, the weights of different neurons would diverge (region marked "training" in Fig. 3A). Our strategy to make them convolutional is to introduce an occasional sleep phase, during which the weights relax to their mean over output neurons (region marked "sleep" in Fig. 3A). This will compensate weight divergence during learning by convergence during the sleep phase. If the latter is sufficiently strong, the weights will remain approximately convolutional.

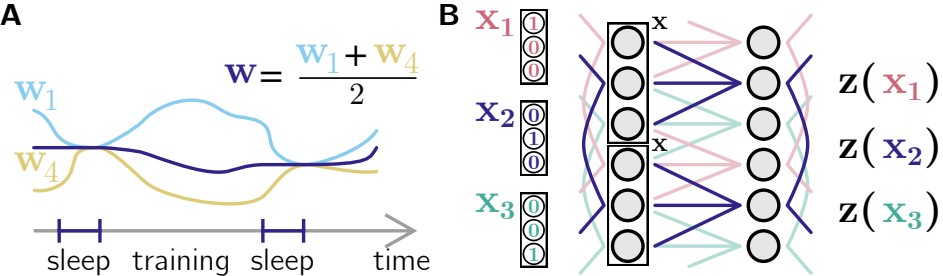

Figure 3: **A.** Dynamical weight sharing interrupts the main training loop, and equalizes the weights through internal dynamics. After that, the weights diverge again until the next weight sharing phase. **B.** A locally connected network, where both the input and the output neurons have lateral connections. The input layer uses lateral connections to generate repeated patterns for weight sharing in the output layer. For instance, the output neurons connected by the dark blue lateral connections (middle) can receive three different patterns: $\mathbf{x}_1$ (generated by the red input grid), $\mathbf{x}_2$ (dark blue) and $\mathbf{x}_3$ (green).

To implement this, we introduce lateral connectivity, chosen to equalize activity in both the input and output layer. That's shown in Fig. 3B, where every third neuron in both the input ($\mathbf{x}$) and output ($\mathbf{z}$) layers are connected. Once the connected neurons in the input layer have equal activity, all output neurons receive identical input. Since the lateral output connections also equalize activity, all connection that correspond to a translation by three neurons see exactly the same pre and postsynaptic activity. A naive Hebbian learning rule (with weight decay) would, therefore, make the network convolutional. However, we have to take care that the initial weighs are not over-written during Hebbian learning. We now describe how that is done.

To ease notation, we'll let $\mathbf{w}_i$ be a vector containing the incoming weights to neuron $i$: $(\mathbf{w}_i)_j \equiv w_{ij}$. Moreover, we'll let $j$ run from 1 to $k$, independent of $i$. With this convention, the response of neuron $i$, $z_i$, to a $k$-dimensional input, $\mathbf{x}$, is given by

$$z_i = \mathbf{w}_i^\top \mathbf{x} = \sum_{j=1}^{k} w_{ij} x_j \,. \tag{3}$$

Assume that every neuron sees the same $\mathbf{x}$, and consider the following update rule for the weights,

$$\Delta \mathbf{w}_i \propto - \left( z_i - \frac{1}{N} \sum_{j=1}^{N} z_j \right) \mathbf{x} - \gamma \left( \mathbf{w}_i - \mathbf{w}_i^{\text{init}} \right) \,, \tag{4}$$

where $\mathbf{w}_i^{\text{init}}$ are the weights at the beginning of the sleep phase (not the overall training).

This Hebbian update effectively implements SGD over the sum of $(z_i - z_j)^2$, plus a regularizer (the second term) to keep the weights near $\mathbf{w}_i^{\text{init}}$. If we present the network with $M$ different input vectors, $\mathbf{x}_m$, and denote the covariance matrix $\mathbf{C} \equiv \frac{1}{M} \sum_m \mathbf{x}_m \mathbf{x}_m^\top$, then, asymptotically, the weight dynamics in Eq. (4) converges to (see Appendix A)

$$\mathbf{w}_i^* = (\mathbf{C} + \gamma \mathbf{I})^{-1} \left( \mathbf{C} \frac{1}{N} \sum_{j=1}^{N} \mathbf{w}_j^{\text{init}} + \gamma \mathbf{w}_i^{\text{init}} \right) \tag{5}$$

where $\mathbf{I}$ is the identity matrix.

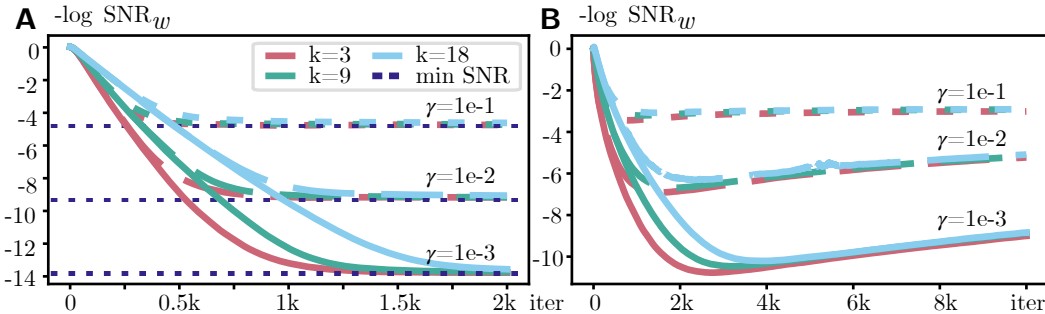

Figure 4: Negative logarithm of signal-to-noise ratio (mean weight squared over weight variance, see Eq. (6)) for weight sharing objectives in a layer with 100 neurons. Different curves have different kernel size, $k$ (meaning $k^2$ inputs), and regularization parameter, $\gamma$. **A.** Weight updates given by Eq. (4). Black dashed lines show the theoretical minimum. **B.** Weight updates given by Eq. (8), with $\alpha = 10$. In each iteration, the input is presented for 150 ms.

As long as **C** is full rank and $\gamma$ is small, we arrive at shared weights: $\mathbf{w}_i^* \approx \frac{1}{N} \sum_{i=1}^{N} \mathbf{w}_i^{\text{init}}$. It might seem advantageous to set $\gamma = 0$, as non-zero $\gamma$ only biases the equilibrium value of the weight. However, non-zero $\gamma$ ensures that for noisy input, $\mathbf{x}_i = \mathbf{x} + \xi_i$ (such that the inputs to different neurons are the same only on average, which is much more realistic), the weights still converge (at least approximately) to the mean of the initial weights (see Appendix A).

In practice, the dynamics in Eq. (4) converges quickly. We illustrate it in Fig. 4A by plotting $-\log \text{SNR}_w$ over time, where $\text{SNR}_w$, the signal to noise ratio of the weights, is defined as

$$\text{SNR}_w = \frac{1}{k^2} \sum_j \frac{\left( \frac{1}{N} \sum_i (\mathbf{w}_i)_j \right)^2}{\frac{1}{N} \sum_i \left( (\mathbf{w}_i)_j - \frac{1}{N} \sum_{i'} (\mathbf{w}_{i'})_j \right)^2} . \tag{6}$$

For all kernel sizes (we used 2d inputs, meaning $k^2$ inputs per neuron), the weights converge to a nearly convolutional solution within a few hundred iterations (note the logarithmic scale of the y axis in Fig. 4A). See Appendix A for simulation details. Thus, to run our experiments with deep networks in a realistic time frame, we perform weight sharing instantly (i.e., directly setting them to the mean value) during the sleep phase.

### 4.1 Dynamic weight sharing in multiple locally connected layers

As shown in Fig. 3B, the $k$-dimensional input, $\mathbf{x}$, repeats every $k$ neurons. Consequently, during the sleep phase, the weights are not set to the mean of their initial value averaged across all neurons; instead, they're set to the mean averaged across a set of neurons spaced by $k$. Thus, in one dimension, the sleep phase equilibrates the weights in $k$ different modules. In two dimensions (the realistic case), the sleep phase equilibrates the weights in $k^2$ different modules.

We need this spacing to span the whole $k$-dimensional (or $k^2$ for 2d) space of inputs. For instance, activating the red grid on the left in Fig. 3B generates $\mathbf{x}_1$, covering one input direction for all output neurons (and within each module, every neuron receives the same input). Next, activating the blue grid generates $\mathbf{x}_2$ (a new direction), and so on.

In multiple layers, the sleep phase is implemented layer by layer. In layer $l$, lateral connectivity creates repeated input patterns and feeds them to layer $l + 1$. After weight sharing in layer $l + 1$, the new pattern from $l + 1$ is fed to $l + 2$, and so on. Notably, there's no layer by layer plasticity schedule (i.e., deeper layers don't have to wait for the earlier ones to finish), as the weight decay term in Eq. (4) ensures the final solution is the same regardless of intermediate weight updates. As long as a deeper layer starts receiving repeated patterns, it will eventually arrive at the correct solution. Our approach has two limitations. First, this pattern generation scheme needs layers to have filters of the same size. Second, we assume that the very first layer (e.g., V1) receives inputs from another area (e.g., LGN) that can generate repeated patterns, but doesn't need weight sharing.

## 4.2 A realistic model that implements the update rule

Our update rule, Eq. (4), implies that there is a linear neuron, denoted $r_i$, whose activity depends on the upstream input, $z_i = \mathbf{w}_i^\top \mathbf{x}$, via a direct excitatory connection combined with lateral inhibition,

$$r_i = z_i - \frac{1}{N} \sum_{j=1}^{N} z_j \equiv \mathbf{w}_i^\top \mathbf{x} - \frac{1}{N} \sum_{j=1}^{N} \mathbf{w}_j^\top \mathbf{x}. \tag{7}$$

The resulting update rule is anti-Hebbian, $-r_i \mathbf{x}$ (see Eq. (4)). In a realistic circuit, this can be implemented with excitatory neurons $r_i$ and an inhibitory neuron $r_{\mathrm{inh}}$, which obey the dynamics

$$\tau \dot{r}_i = -r_i + \mathbf{w}_i^\top \mathbf{x} - \alpha \, r_{\mathrm{inh}} + b \tag{8a}$$

$$\tau \dot{r}_{\mathrm{inh}} = -r_{\mathrm{inh}} + \frac{1}{N} \sum_j r_j - b, \tag{8b}$$

where $b$ is the shared bias term that ensures non-negativity of firing rates (assuming $\sum_i \mathbf{w}_i^\top \mathbf{x}$ is positive, which would be the case for excitatory input neurons). The only fixed point of this equations is

$$r_i^* = b + \mathbf{w}_i^\top \mathbf{x} - \frac{1}{N} \sum_j \mathbf{w}_j^\top \mathbf{x} + \frac{1}{1+\alpha} \frac{1}{N} \sum_j \mathbf{w}_j^\top \mathbf{x} \underset{\alpha \gg 1}{\approx} b + \mathbf{w}_i^\top \mathbf{x} - \frac{1}{N} \sum_j \mathbf{w}_j^\top \mathbf{x}, \tag{9}$$

which is stable. As a result, for strong inhibition ($\alpha \gg 1$), Eq. (4) can be implemented with an anti-Hebbian term $-(r_i - b)\mathbf{x}$. Note that if $\mathbf{w}_i^\top \mathbf{x}$ is zero on average, then $b$ is the mean firing rate over time. To show that Eq. (9) provides enough signal, we simulated training in a network of 100 neurons that receives a new $\mathbf{x}$ each 150 ms. For a range of $k$ and $\gamma$, it converged to a nearly convolutions solution within minutes (Fig. 4B; each iteration is 150 ms). Having finite inhibition did lead to worse final signal-to-noise ration ($\alpha = 10$ in Fig. 4B), but the variance of the weights was still very small. Moreover, the nature of the $\alpha$-induced bias suggests that stopping training before convergence leads to better results (around 2k iterations in Fig. 4B). See Appendix A for a discussion.

## 5 Experiments

We split our experiments into two parts: small-scale ones with CIFAR10, CIFAR100 [35] and TinyImageNet [36], and large-scale ones with ImageNet [37]. The former illustrates the effects of data augmentation and dynamic weight sharing on the performance of locally connected networks; the latter concentrates on dynamic weight sharing, as extensive data augmentations are too computationally expensive for large networks and datasets. We used the AdamW [38] optimizer in all runs. As our dynamic weight sharing procedure always converges to a nearly convolutional solution (see Section 4), we set the weights to the mean directly (within each grid) to speed up experiments. Our code is available at https://github.com/romanpogodin/towards-bio-plausible-conv (PyTorch [39] implementation).

**Datasets.** CIFAR10 consists of 50k training and 10k test images of size $32 \times 32$, divided into 10 classes. CIFAR100 has the same structure, but with 100 classes. For both, we tune hyperparameters with a 45k/5k train/validation split, and train final networks on the full 50k training set. TinyImageNet consists of 100k training and 10k validation images of size $64 \times 64$, divided into 200 classes. As the test labels are not publicly available, we divided the training set into 90k/10k train/validation split, and used the 10k official validation set as test data. ImageNet consists of 1.281 million training images and 50k test images of different sizes, reshaped to 256 pixels in the smallest dimension. As in the case for TinyImageNet, we used the train set for a 1.271 million/10k train/validation split, and 50k official validation set as test data.

**Networks.** For CIFAR10/100 and TinyImageNet, we used CIFAR10-adapted ResNet20 from the original ResNet paper [3]. The network has three blocks, each consisting of 6 layers, with 16/32/64 channels within the block. We chose this network due to good performance on CIFAR10, and the ability to fit the corresponding locally connected network into the 8G VRAM of the GPU for large batch sizes on all three datasets. For ImageNet, we took the half-width ResNet18 (meaning 32/64/128/256 block widths) to be able to fit a common architecture (albeit halved in width) in the locally connected regime into 16G of GPU VRAM. For both networks, all layers had $3 \times 3$ receptive

Table 1: Performance of convolutional (conv) and locally connected (LC) networks for padding of 4 in the input images (mean accuracy over 5 runs). For LC, two regularization strategies were applied: repeating the same image $n$ times with different translations ($n$ reps) or using dynamic weight sharing every $n$ batches (ws($n$)). LC nets additionally show performance difference w.r.t. conv nets.

| Regularizer | Connectivity | CIFAR10 | | CIFAR100 | | | | TinyImageNet | | | |
| | | Top-1 accuracy (%) | Diff | Top-1 accuracy (%) | Diff | Top-5 accuracy (%) | Diff | Top-1 accuracy (%) | Diff | Top-5 accuracy (%) | Diff |
|---|---|---|---|---|---|---|---|---|---|---|---|
| - | conv | 88.3 | - | 59.2 | - | 84.9 | - | 38.6 | - | 65.1 | - |
| | LC | 80.9 | -7.4 | 49.8 | -9.4 | 75.5 | -9.4 | 29.6 | -9.0 | 52.7 | -12.4 |
| Data Translation | LC - 4 reps | 82.9 | -5.4 | 52.1 | -7.1 | 76.4 | -8.5 | 31.9 | -6.7 | 54.9 | -10.2 |
| | LC - 8 reps | 83.8 | -4.5 | 54.3 | -5.0 | 77.9 | -7.0 | 33.0 | -5.6 | 55.6 | -9.5 |
| | LC - 16 reps | 85.0 | -3.3 | 55.9 | -3.3 | 78.8 | -6.1 | 34.0 | -4.6 | 56.2 | -8.8 |
| Weight Sharing | LC - ws(1) | 87.4 | -0.8 | 58.7 | -0.5 | 83.4 | -1.6 | 41.6 | 3.0 | 66.1 | 1.1 |
| | LC - ws(10) | 85.1 | -3.2 | 55.7 | -3.6 | 80.9 | -4.0 | 37.4 | -1.2 | 61.8 | -3.2 |
| | LC - ws(100) | 82.0 | -6.3 | 52.8 | -6.4 | 80.1 | -4.8 | 37.1 | -1.5 | 62.8 | -2.3 |

field (apart from a few $1 \times 1$ residual downsampling layers), meaning that weight sharing worked over 9 individual grids in each layer.

**Training details.** We ran the experiments on our local laboratory cluster, which consists mostly of NVIDIA GTX1080 and RTX5000 GPUs. The small-scale experiments took from 1-2 hours per run up to 40 hours (for TinyImageNet with 16 repetitions). The large-scale experiments took from 3 to 6 days on RTX5000 (the longest run was the locally connected network with weight sharing happening after every minibatch update).

## 5.1 Data augmentations.

For CIFAR10/100, we padded the images (padding size depended on the experiment) with mean values over the training set (such that after normalization the padded values were zero) and cropped to size $32 \times 32$. We did not use other augmentations to separate the influence of padding/random crops. For TinyImageNet, we first center-cropped the original images to size $(48 + 2\,\mathrm{pad}) \times (48 + 2\,\mathrm{pad})$ for the chosen padding size pad. The final images were then randomly cropped to $48 \times 48$. This was done to simulate the effect of padding on the number of available translations, and to compare performance across different padding values on the images of the same size (and therefore locally connected networks of the same size). After cropping, the images were normalized using ImageNet normalization values. For all three datasets, test data was sampled without padding. For ImageNet, we used the standard augmentations. Training data was resized to 256 (smallest dimension), random cropped to $224 \times 224$, flipped horizontally with $0.5$ probability, and then normalized. Test data was resized to 256, center cropped to 224 and then normalized. In all cases, data repetitions included multiple samples of the same image within a batch, keeping the total number of images in a batch fixed (e.g. for batch size 256 and 16 repetitions, that would mean 16 original images)

## 5.2 CIFAR10/100 and TinyImageNet

To study the effect of both data augmentation and weight sharing on performance, we ran experiments with non-augmented images (padding 0) and with different amounts of augmentations. This included padding of 4 and 8, and repetitions of 4, 8, and 16. Without augmentations, locally connected networks performed much worse than convolutional, although weight sharing improved the result a little bit (see Appendix B). For padding of 4 (mean accuracy over 5 runs Table 1, see Appendix B for max-min accuracy), increasing the number of repetitions increased the performance of locally connected networks. However, even for 16 repetitions, the improvements were small comparing to weight sharing (especially for top-5 accuracy on TinyImageNet). For CIFAR10, our results are consistent with an earlier study of data augmentations in locally connected networks [40]. For dynamic weight sharing, doing it moderately often – every 10 iterations, meaning every 5120 images – did as well as 16 repetitions on CIFAR10/100. For TinyImageNet, sharing weights every 100 iterations (about every 50k images) performed much better than data augmentation.

Sharing weights after every batch performed almost as well as convolutions (and even a bit better on TinyImageNet, although the difference is small if we look at top-5 accuracy, which is a less volatile metric for 200 classes), but it is too frequent to be a plausible sleep phase. We include it to show that best possible performance of partial weight sharing is comparable to actual convolutions.

Table 2: Performance of convolutional (conv) and locally connected (LC) networks on ImageNet for 0.5x width ResNet18 (1 run). For LC, we also used dynamic weight sharing every $n$ batches. LC nets additionally show performance difference w.r.t. the conv net.

| Connectivity | Weight sharing frequency | ImageNet | | | |
|---|---|---|---|---|---|
| | | Top-1 accuracy (%) | Diff | Top-5 accuracy (%) | Diff |
| conv | - | 63.5 | - | 84.7 | - |
| LC | - | 46.7 | -16.8 | 70.0 | -14.7 |
| LC | 1 | 61.7 | -1.8 | 83.1 | -1.6 |
| LC | 10 | 59.3 | -4.2 | 81.1 | -3.6 |
| LC | 100 | 54.5 | -9.0 | 77.7 | -7.0 |

For a padding of 8, the performance did improve for all methods (including convolutions), but the relative differences had a similar trend as for a padding of 4 (see Appendix B). We also trained locally connected networks with one repetition, but for longer and with a much smaller learning rate to simulate the effect of data repetitions. Even for 4x-8x longer runs, the networks barely matched the performance of a 1-repetition network on standard speed (not shown).

## 5.3 ImageNet

On ImageNet, we did not test image repetitions due to the computational requirements (e.g., running 16 repetitions with our resources would take almost 3 months). We used the standard data augmentation, meaning that all networks see different crops of the same image throughout training.

Our results are shown in Table 2. Weight sharing every 1 and 10 iterations (256/2560 images, respectively, for the batch size of 256) achieves nearly convolutional performance, although less frequent weight sharing results in a more significant performance drop. In contrast, the purely locally connected network has a large performance gap with respect to the convolutional one. It is worth noting that the trade-off between weight sharing frequency and performance depends on the learning rate, as weights diverge less for smaller learning rates. It should be possible to decrease the learning rate and increase the number of training epochs, and achieve comparable results with less frequent weight sharing.

## 5.4 Brain-Score of ImageNet-trained networks

In addition to ImageNet performance, we evaluated how well representations built by our networks correspond to ventral stream data in primates. For that we used the Brain-Score [12, 17], a set of metrics that evaluate deep networks' correspondence to neural recordings of cortical areas V1 [41, 42], V2 [41], V4 [43], and inferior temporal (IT) cortex [43] in primates, as well as behavioral data [44]. The advantage of Brain-Score is that it provides a standardized benchmark that looks at the whole ventral stream, and not only its isolated properties like translation invariance (which many early models focused on [45, 46, 47, 48]). We do not directly check for translation invariance in our models (only through V1/V2 data). However, as our approach achieves convolutional solutions (see above), we trivially have translation equivariance after training: translating the input will translate the layer's response (in our case, each $k$-th translation will produce the same translated response for a kernel of size $k$). (In fact, it's pooling layers, not convolutions, that achieve some degree of invariance in convolutional networks – this is an architectural choice and is somewhat tangential to our problem.)

Our results are shown in Table 3 (higher is better; see http://www.brain-score.org/ for the scores of other models). The well-performing locally connected networks (with weight sharing/sleep phase every 1 or 10 iterations) show overall better fit compared to their fully convolutional counterpart – meaning that plausible approximations to convolutions also form more realistic representations. Interestingly, the worst-performing purely locally connected network had the second best V1 fit, despite overall poor performance.

In general, Brain-Score correlates with ImageNet performance (Fig. 2 in [12]). This means that the worse Brain-Score performance of the standard locally connected network and the one with weight sharing every 100 iterations can be related to their poor ImageNet performance (Table 2). (It also means that our method can potentially increase Brain-Score performance of larger models.)

Table 3: Brain-Score of ImageNet-trained convolutional (conv) and locally connected (LC) networks on ImageNet for 0.5x width ResNet18 (higher is better). For LC, we also used dynamic weight sharing every $n$ batches. *The models were evaluated on Brain-Score benchmarks available during submission. If new benchmarks are added and the models are re-evaluated on them, the final scores might change; the provided links contain the latest results.

| Connectivity | Weight sharing frequency | Brain-Score | | | | | | |
|---|---|---|---|---|---|---|---|---|
| | | average score | V1 | V2 | V4 | IT | behavior | link* |
| conv | - | .357 | .493 | .313 | .459 | .370 | .148 | brain-score.org/model/876 |
| LC | - | .349 | **.542** | .291 | .448 | .354 | .108 | brain-score.org/model/877 |
| LC | 1 | **.396** | .512 | **.339** | .468 | **.406** | **.255** | brain-score.org/model/880 |
| LC | 10 | .385 | .508 | .322 | **.478** | .399 | .216 | brain-score.org/model/878 |
| LC | 100 | .351 | .523 | .293 | .467 | .370 | .101 | brain-score.org/model/879 |

## 6 Discussion

We presented two ways to circumvent the biological implausibility of weight sharing, a crucial component of convolutional networks. The first was through data augmentation via multiple image translations. The second was dynamic weight sharing via lateral connections, which allows neurons to share weight information during a sleep-like phase; weight updates are then done using Hebbian plasticity. Data augmentation requires a large number of repetitions in the data, and, consequently, longer training times, and yields only small improvements in performance. However, only a small number of repetitions can be naturally covered by saccades. Dynamic weight sharing needs a separate sleep phase, rather than more data, and yields large performance gains. In fact, it achieves near convolutional performance even on hard tasks, such as ImageNet classification, making it a much more likely candidate than data augmentation for the brain. In addition, well-performing locally connected networks trained with dynamic weight sharing achieve a better fit to the ventral stream data (measured by the Brain-Score [12, 17]). The sleep phase can occur during actual sleep, when the network (i.e., the visual system) stops receiving visual inputs, but maintains some internal activity. This is supported by plasticity studies during sleep (e.g. [49, 50]).

There are several limitations to our implementation of dynamic weight sharing. First, it relies on precise lateral connectivity. This can be genetically encoded, or learned early on using correlations in the input data (if layer $l$ can generate repeated patters, layer $l + 1$ can modify its lateral connectivity based on input correlations). Lateral connections do in fact exist in the visual stream, with neurons that have similar tuning curves showing strong lateral connections [51]. Second, the sleep phase works iteratively over layers. This can be implemented with neuromodulation that enables plasticity one layer at a time. Alternatively, weight sharing could work simultaneously in the whole network due to weight regularization (as it ensures that the final solution preserves the initial average weight), although this would require longer training due to additional noise in deep layers. Third, in our scheme the lateral connections are used only for dynamic weight sharing, and not for training or inference. As our realistic model in Section 4.2 implements this connectivity via an inhibitory neuron, we can think of that neuron as being silent outside of the sleep phase. Finally, we trained networks using backpropagation, which is not biologically plausible [18]. However, our weight sharing scheme is independent of the wake-phase training algorithm, and therefore can be applied along with any biologically plausible update rule.

Our approach to dynamic weight sharing is not relevant only to convolutions. First, it is applicable to non-convolutional networks, and in particular visual transformers [28, 29, 30, 31] (and more recent MLP-based architectures [32, 33, 34]). In such architectures, input images (and intermediate two-dimensional representations) are split into non-overlapping patches; each patch is then transformed with the *same* fully connected layer – a computation that would require weight sharing in the brain. This can be done by connecting neurons across patches that have the same relative position, and applying our weight dynamics (see Appendix A.4). Second, [23] faced a problem similar to weight sharing – weight transport (i.e., neurons not knowing their output weights) – when developing a plausible implementation of backprop. Their weight mirror algorithms used an idea similar to ours: the value of one weight was sent to another through correlations in activity.

Our study shows that both performance and the computation of convolutional networks can be reproduced in more realistic architectures. This supports convolutional networks as a model of the

visual stream, and also justifies them as a "model organism" for studying learning in the visual stream (which is important partially due to their computational efficiency). While our study does not have immediate societal impacts (positive or negative), it further strengthens the role of artificial neural networks as a model of the brain. Such models can guide medical applications such as brain machine interfaces and neurological rehabilitation. However, that could also lead to the design of potentially harmful adversarial attacks on the brain.

## Acknowledgments and Disclosure of Funding

The authors would like to thank Martin Schrimpf and Mike Ferguson for their help with the Brain-Score evaluation.

This work was supported by the Gatsby Charitable Foundation, the Wellcome Trust and DeepMind.

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
