# Appendices

## A  Dynamic weight sharing

### A.1  Noiseless case

Each neuron receives the same $k$-dimensional input $\mathbf{x}$, and its response $z_i$ is given by

$$z_i = \mathbf{w}_i^\top \mathbf{x} = \sum_{j=1}^{k} w_{ij} x_j \, . \tag{10}$$

To equalize the weights $\mathbf{w}_i$ among all neurons, the network minimizes the following objective,

$$\mathcal{L}_{\text{w. sh.}}(\mathbf{w}_1, \ldots, \mathbf{w}_N) = \frac{1}{4MN} \sum_{m=1}^{M} \sum_{i=1}^{N} \sum_{j=1}^{N} (z_i - z_j)^2 + \frac{\gamma}{2} \sum_{i=1}^{N} \|\mathbf{w}_i - \mathbf{w}_i^{\text{init}}\|^2 \tag{11}$$

$$= \frac{1}{4MN} \sum_{m=1}^{M} \sum_{i=1}^{N} \sum_{j=1}^{N} \left( \mathbf{w}_i^\top \mathbf{x}_m - \mathbf{w}_j^\top \mathbf{x}_m \right)^2 + \frac{\gamma}{2} \sum_{i=1}^{N} \|\mathbf{w}_i - \mathbf{w}_i^{\text{init}}\|^2 \, , \tag{12}$$

where $\mathbf{w}_i^{\text{init}}$ is the weight at the start of dynamic weight sharing. This is a strongly convex function, and therefore it has a unique minimum.

The SGD update for one $\mathbf{x}_m$ is

$$\Delta \mathbf{w}_i \propto - \left( z_i - \frac{1}{N} \sum_{j=1}^{N} z_j \right) \mathbf{x}_m - \gamma \left( \mathbf{w}_i - \mathbf{w}_i^{\text{init}} \right) \, . \tag{13}$$

To find the fixed point of the dynamics, we first set the sum over the gradients to zero,

$$\sum_i \frac{d\,\mathcal{L}_{\text{w. sh.}}(\mathbf{w}_1, \ldots, \mathbf{w}_N)}{d\,\mathbf{w}_i} = \frac{1}{M} \sum_{i,m} \left( z_i - \frac{1}{N} \sum_{j=1}^{N} z_j \right) \mathbf{x}_m + \gamma \sum_i \left( \mathbf{w}_i - \mathbf{w}_i^{\text{init}} \right) \tag{14}$$

$$= \gamma \sum_i \left( \mathbf{w}_i - \mathbf{w}_i^{\text{init}} \right) = 0 \, . \tag{15}$$

Therefore, at the fixed point the mean weight $\boldsymbol{\mu}^* = \sum_i \mathbf{w}_i^* / N$ is equal to $\boldsymbol{\mu}^{\text{init}} = \sum_i \mathbf{w}_i^{\text{init}} / N$, and

$$\frac{1}{N} \sum_{i=1}^{N} z_i = \frac{1}{N} \sum_{i=1}^{N} \mathbf{w}_i^{*\top} \mathbf{x}_m = (\boldsymbol{\mu}^{\text{init}})^\top \mathbf{x}_m \, . \tag{16}$$

We can now find the individual weights,

$$\frac{d\,\mathcal{L}_{\text{w. sh.}}(\mathbf{w}_1, \ldots, \mathbf{w}_N)}{d\,\mathbf{w}_i} = \frac{1}{M} \sum_m \left( z_i - \frac{1}{N} \sum_{j=1}^{N} z_j \right) \mathbf{x}_m + \gamma \left( \mathbf{w}_i - \mathbf{w}_i^{\text{init}} \right) \tag{17}$$

$$= \frac{1}{M} \sum_m \mathbf{x}_m \mathbf{x}_m^\top \left( \mathbf{w}_i - \boldsymbol{\mu}^{\text{init}} \right) + \gamma \left( \mathbf{w}_i - \mathbf{w}_i^{\text{init}} \right) = 0 \, . \tag{18}$$

Denoting the covariance matrix $\mathbf{C} \equiv \frac{1}{M} \sum_m \mathbf{x}_m \mathbf{x}_m^\top$, we see that

$$\mathbf{w}_i^* = (\mathbf{C} + \gamma \mathbf{I})^{-1} \left( \mathbf{C} \boldsymbol{\mu}^{\text{init}} + \gamma \mathbf{w}_i^{\text{init}} \right) = (\mathbf{C} + \gamma \mathbf{I})^{-1} \left( \mathbf{C} \frac{1}{N} \sum_{i=1}^{N} \mathbf{w}_i^{\text{init}} + \gamma \mathbf{w}_i^{\text{init}} \right) , \tag{19}$$

where $\mathbf{I}$ is the identity matrix. From Eq. (19) it is clear that $\mathbf{w}_i^* \approx \boldsymbol{\mu}^{\text{init}}$ for small $\gamma$ and full rank $\mathbf{C}$. For instance, for $\mathbf{C} = \mathbf{I}$,

$$\mathbf{w}_i^* = \frac{1}{1+\gamma} \boldsymbol{\mu}^{\text{init}} + \frac{\gamma}{1+\gamma} \mathbf{w}_i^{\text{init}} \, . \tag{20}$$

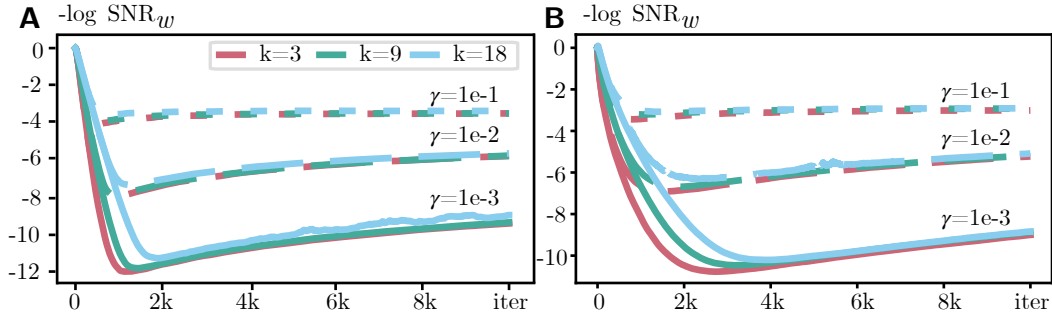

Figure 5: Logarithm of inverse signal-to-noise ratio (mean weight squared over weight variance, see Eq. (6)) for weight sharing objectives in a layer with 100 neurons. **A.** Dynamics of Eq. (21) for different kernel sizes $k$ (meaning $k^2$ inputs) and $\gamma$. **B.** Dynamics of weight update that uses Eq. (8b) for $\alpha = 10$, different kernel sizes $k$ and $\gamma$. In each iteration, the input is presented for 150 ms.

## A.2 Biased noiseless case, and its correspondence to the realistic implementation

The realistic implementation of dynamic weight sharing with an inhibitory neuron (Section 4.2) introduces a bias in the update rule: Eq. (13) becomes

$$\Delta \mathbf{w}_i \propto - \left( z_i - \frac{\alpha}{N(1+\alpha)} \sum_{j=1}^{N} z_j \right) \mathbf{x}_m - \gamma \left( \mathbf{w}_i - \mathbf{w}_i^{\text{init}} \right) \quad (21)$$

for inhibition strength $\alpha$.

Following the same derivation as for the unbiased case, we can show that the weight dynamics converges to

$$\sum_i \frac{d \mathcal{L}_{\text{w. sh.}}(\mathbf{w}_1, \ldots, \mathbf{w}_N)}{d \mathbf{w}_i} = \frac{1}{M} \sum_{i,m} \left( z_i - \frac{\alpha}{1+\alpha} \frac{1}{N} \sum_{j=1}^{N} z_j \right) \mathbf{x}_m + \gamma \sum_i \left( \mathbf{w}_i - \mathbf{w}_i^{\text{init}} \right) \quad (22)$$

$$= \frac{1}{1+\alpha} \mathbf{C} \sum_i \mathbf{w}_i + \gamma \sum_i \left( \mathbf{w}_i - \mathbf{w}_i^{\text{init}} \right) = 0. \quad (23)$$

Therefore $\boldsymbol{\mu}^* = \gamma \left( \frac{1}{1+\alpha} \mathbf{C} + \gamma \mathbf{I} \right)^{-1} \boldsymbol{\mu}^{\text{init}}$, and

$$\mathbf{w}_i^* = (\mathbf{C} + \gamma \mathbf{I})^{-1} \left( \frac{\gamma \alpha}{1+\alpha} \mathbf{C} \left( \frac{1}{1+\alpha} \mathbf{C} + \gamma \mathbf{I} \right)^{-1} \boldsymbol{\mu}^{\text{init}} + \gamma \mathbf{w}_i^{\text{init}} \right). \quad (24)$$

For $\mathbf{C} = \mathbf{I}$, this becomes

$$\mathbf{w}_i^* = \frac{\gamma}{1+\gamma} \left( \frac{\alpha}{1+\gamma(1+\alpha)} \boldsymbol{\mu}^{\text{init}} + \mathbf{w}_i^{\text{init}} \right). \quad (25)$$

As a result, the final weights are approximately the same among neurons, but have a small norm due to the $\gamma$ scaling.

The dynamics in Eq. (21) correctly captures the bias influence in Eq. (8b), producing similar SNR plots; compare Fig. 5A (Eq. (21) dynamics) to Fig. 5B (Eq. (8b) dynamics). The curves are slightly different due to different learning rates, but both follow the same trend of first finding a very good solution, and then slowly incorporating the bias term (leading to smaller SNR).

## A.3 Noisy case

Realistically, all neurons can't see the same $\mathbf{x}_m$. However, due to the properties of our loss, we can work even with noisy updates. To see this, we write the objective function as

$$\mathcal{L}_{\text{w. sh.}}(\mathbf{w}_1, \ldots, \mathbf{w}_N) = \frac{1}{M} \sum_{m=1}^{M} f(\mathbf{W}, \mathbf{X}_m) \tag{26}$$

where matrices $\mathbf{W}$ and $\mathbf{X}$ satisfy $(\mathbf{W})_i = \mathbf{w}_i$ and $(\mathbf{X}_m)_i = \mathbf{x}_m$, and

$$f(\mathbf{W}, \mathbf{X}_m) = \frac{1}{4N} \sum_{i=1}^{N} \sum_{j=1}^{N} \left( \mathbf{w}_i^\top \mathbf{x}_m - \mathbf{w}_i^\top \mathbf{x}_m \right)^2 + \frac{\gamma}{2} \sum_{i=1}^{N} \left\| \mathbf{w}_i - \mathbf{w}_i^{\text{init}} \right\|^2. \tag{27}$$

We'll update the weights with SGD according to

$$\Delta \mathbf{W}^{k+1} = -\eta_k \left. \frac{d}{d \mathbf{W}} f(\mathbf{W}, \mathbf{X}_{m(k)} + \mathbf{E}^k) \right|_{\mathbf{W}^k}, \tag{28}$$

where $(\mathbf{E}^k)_i = \boldsymbol{\epsilon}_i$ is zero-mean input noise and $m(k)$ is chosen uniformly.

Let's also bound the input mean and noise as

$$\mathbb{E}_{\mathbf{E}} \left\| \mathbf{x}_{m(k)} + \boldsymbol{\epsilon}_i \right\|^2 \leq \sqrt{c_{x\epsilon}}, \quad \mathbb{E}_{\mathbf{E}} \left\| \mathbf{x}_{m(k)} + \boldsymbol{\epsilon}_i \right\|^4 \leq c_{x\epsilon}. \tag{29}$$

With this setup, we can show that SGD with noise can quickly converge to the correct solution, apart from a constant noise-induced bias. Our analysis is standard and follows [52], but had to be adapted for our objective and noise model.

**Theorem 1.** *For zero-mean isotropic noise $\mathbf{E}$ with variance $\sigma^2$, uniform SGD sampling $m(k)$ and inputs $\mathbf{x}_m$ that satisfy Eq. (29), choosing $\eta_k = O(1/k)$ leads to*

$$\mathbb{E} \left\| \mathbf{W}^{k+1} - \mathbf{W}^* \right\|_F^2 = O \left( \frac{\left\| \mathbf{W}^{\text{init}} - \mathbf{W}^* \right\|_F}{k+1} \right) + O \left( \sigma^2 \| \mathbf{W}^* \|_F^2 \right), \tag{30}$$

*where $(\mathbf{W}^*)_i$ is given by Eq. (19).*

*Proof.* Using the SGD update,

$$\left\| \mathbf{W}^{k+1} - \mathbf{W}^* \right\|_F^2 = \left\| \mathbf{W}^k - \eta_k \left. \frac{d}{d \mathbf{W}} f(\mathbf{W}, \mathbf{X}_{m(k)} + \mathbf{E}^k) \right|_{\mathbf{W}^k} - \mathbf{W}^* \right\|_F^2 \tag{31}$$

$$= \left\| \mathbf{W}^k - \mathbf{W}^* \right\|_F^2 - 2\eta_k \left\langle \mathbf{W}^k - \mathbf{W}^*, \left. \frac{d}{d \mathbf{W}} f(\mathbf{W}, \mathbf{X}_{m(k)} + \mathbf{E}^k) \right|_{\mathbf{W}^k} \right\rangle \tag{32}$$

$$+ \eta_k^2 \left\| \left. \frac{d}{d \mathbf{W}} f(\mathbf{W}, \mathbf{X}_{m(k)} + \mathbf{E}^k) \right|_{\mathbf{W}^k} \right\|_F^2. \tag{33}$$

We need to bound the second and the third terms in the equation above.

**Second term.** As $f$ is $\gamma$-strongly convex in $\mathbf{W}$,

$$- \left\langle \mathbf{W}^k - \mathbf{W}^*, \left. \frac{d}{d \mathbf{W}} f(\mathbf{W}, \mathbf{X}_{m(k)} + \mathbf{E}^k) \right|_{\mathbf{W}^k} \right\rangle \tag{34}$$

$$\leq f(\mathbf{W}^*, \mathbf{X}_{m(k)} + \mathbf{E}^k) - f(\mathbf{W}^k, \mathbf{X}_{m(k)} + \mathbf{E}^k) - \frac{\gamma}{2} \left\| \mathbf{W}^k - \mathbf{W}^* \right\|_F^2. \tag{35}$$

As $f$ is convex in $\mathbf{X}$,

$$f(\mathbf{W}^*, \mathbf{X}_{m(k)} + \mathbf{E}^k) - f(\mathbf{W}^k, \mathbf{X}_{m(k)} + \mathbf{E}^k) \leq f(\mathbf{W}^*, \mathbf{X}_{m(k)}) - f(\mathbf{W}^k, \mathbf{X}_{m(k)}) \tag{36}$$

$$+ \left\langle \left. \frac{d}{d \mathbf{X}} f(\mathbf{W}^*, \mathbf{X}) \right|_{\mathbf{X}_{m(k)} + \mathbf{E}^k} - \left. \frac{d}{d \mathbf{X}} f(\mathbf{W}^k, \mathbf{X}) \right|_{\mathbf{X}_{m(k)}}, \mathbf{E}^k \right\rangle. \tag{37}$$

We only need to clarify one term here,

$$\left(\left.\frac{d}{d\mathbf{X}}f(\mathbf{W}^*,\mathbf{X})\right|_{\mathbf{X}_{m(k)}+\mathbf{E}^k}\right)_i = \left(\left.\frac{d}{d\mathbf{X}}f(\mathbf{W}^*,\mathbf{X})\right|_{\mathbf{X}_{m(k)}}\right)_i + \left(\mathbf{w}_i^{*\top}\boldsymbol{\epsilon}_i - \frac{1}{N}\sum_j \mathbf{w}_j^{*\top}\boldsymbol{\epsilon}_j\right)\mathbf{w}_i^*. \tag{38}$$

Now we can take the expectation over $m(k)$ and $\mathbf{E}$. As $m(k)$ is uniform, and $\mathbf{W}^*$ minimizes the global function,

$$\mathbb{E}_{m(k)}\left(f(\mathbf{W}^*,\mathbf{X}_{m(k)}) - f(\mathbf{W}^k,\mathbf{X}_{m(k)})\right) = \mathcal{L}_{\text{w. sh.}}(\mathbf{w}_1^*,\dots,\mathbf{w}_N^*) - \mathcal{L}_{\text{w. sh.}}(\mathbf{w}_1^k,\dots,\mathbf{w}_N^k) \le 0. \tag{39}$$

As $\mathbf{E}^k$ is zero-mean and isotropic with variance $\sigma^2$,

$$\mathbb{E}_{m(k),\mathbf{E}^k}\left\langle \left.\frac{d}{d\mathbf{X}}f(\mathbf{W}^*,\mathbf{X})\right|_{\mathbf{X}_{m(k)}+\mathbf{E}^k} - \left.\frac{d}{d\mathbf{X}}f(\mathbf{W}^k,\mathbf{X})\right|_{\mathbf{X}_{m(k)}}, \mathbf{E}^k\right\rangle \tag{40}$$

$$= \mathbb{E}_{\mathbf{E}^k}\sum_i \left(\mathbf{w}_i^{*\top}\boldsymbol{\epsilon}_i - \frac{1}{N}\sum_j \mathbf{w}_j^{*\top}\boldsymbol{\epsilon}_j\right)\mathbf{w}_i^{*\top}\boldsymbol{\epsilon}_i = \left(1-\frac{1}{N}\right)\mathbb{E}_{\mathbf{E}^k}\sum_i \left(\mathbf{w}_i^{*\top}\boldsymbol{\epsilon}_i\right)^2 \tag{41}$$

$$= \left(1-\frac{1}{N}\right)\mathbb{E}_{\mathbf{E}^k}\sum_i \text{Tr}\left(\mathbf{w}_i^*\mathbf{w}_i^{*\top}\boldsymbol{\epsilon}_i\boldsymbol{\epsilon}_i^\top\right) \le \sigma^2\|\mathbf{W}^*\|_F^2. \tag{42}$$

So the whole second term becomes

$$-2\eta_k\mathbb{E}_{m(k),\mathbf{E}}\left\langle \mathbf{W}^k - \mathbf{W}^*, \left.\frac{d}{d\mathbf{W}}f(\mathbf{W},\mathbf{X}_{m(k)}+\mathbf{E}^k)\right|_{\mathbf{W}^k}\right\rangle \tag{43}$$

$$\le -\gamma\eta_k\mathbb{E}_{m(k),\mathbf{E}^k}\|\mathbf{W}^k - \mathbf{W}^*\|_F^2 + \eta_k\sigma^2\|\mathbf{W}^*\|_F^2. \tag{44}$$

**Third term.** First, observe that

$$\frac{d}{d\mathbf{w}_i}f(\mathbf{W},\mathbf{X}) = \mathbf{x}_i\mathbf{x}_i^\top\mathbf{w}_i - \mathbf{x}_i\frac{1}{N}\sum_j \mathbf{x}_j^\top\mathbf{w}_j + \gamma\mathbf{w}_i - \gamma\mathbf{w}_i^{\text{init}} \tag{45}$$

$$= \left(1-\frac{1}{N}\right)\mathbf{A}_i\mathbf{w}_i - \mathbf{B}_i\mathbf{W} + \gamma\mathbf{w}_i - \gamma\mathbf{w}_i^{\text{init}}, \tag{46}$$

where $\mathbf{A}_i = \mathbf{x}_i\mathbf{x}_i^\top$ and $(\mathbf{B}_i)_j = \mathbb{I}[i\ne j]\mathbf{x}_i\mathbf{x}_j^\top/N$.

Therefore, using $\|a+b\|^2 \le 2\|a\|^2 + 2\|b\|^2$ twice, properties of the matrix 2-norm, and $(1-1/N)\le 1$,

$$\left\|\frac{d}{d\mathbf{w}_i}f(\mathbf{W},\mathbf{X})\right\|^2 \le 4\|\mathbf{A}_i\|_2^2\|\mathbf{w}_i\|^2 + 4\|\mathbf{B}_i\|_2^2\|\mathbf{W}\|^2 + 4\gamma^2\|\mathbf{w}_i\|^2 + 4\gamma^2\|\mathbf{w}_i^{\text{init}}\|^2. \tag{47}$$

In our particular case, bounding the 2 norm with the Frobenius norm gives

$$\mathbb{E}_{m(k),\mathbf{E}}\|\mathbf{A}_i\|_2^2 \le \mathbb{E}_{m(k),\mathbf{E}}\left\|(\mathbf{x}_{m(k)}+\boldsymbol{\epsilon}_i)(\mathbf{x}_{m(k)}+\boldsymbol{\epsilon}_i)^\top\right\|_F^2 \tag{48}$$

$$= \mathbb{E}_{m(k),\mathbf{E}}\left\|\mathbf{x}_{m(k)}+\boldsymbol{\epsilon}_i\right\|^4 \le c_{x\epsilon}. \tag{49}$$

Similarly,

$$\mathbb{E}_{m(k),\mathbf{E}}\|\mathbf{B}_i\|_2^2 \le \mathbb{E}_{m(k),\mathbf{E}}\|\mathbf{B}_i\|_F^2 \le \frac{1}{N^2}\mathbb{E}_{m(k),\mathbf{E}}\sum_{j\ne i}\left\|\mathbf{x}_{m(k)}+\boldsymbol{\epsilon}_i\right\|^2\left\|\mathbf{x}_{m(k)}+\boldsymbol{\epsilon}_j\right\|^2 \le \frac{c_{x\epsilon}}{N}. \tag{50}$$

Therefore, we can bound the full gradient by the sum of individual bounds (as it's the Frobenius norm) and using $\|a+b\|^2 \le 2\|a\|^2 + 2\|b\|^2$ again,

$$\mathbb{E}_{m(k),\mathbf{E}}\left\|\left.\frac{d}{d\mathbf{W}}f(\mathbf{W},\mathbf{X}_{m(k)+\mathbf{E}^k})\right|_{\mathbf{W}^k}\right\|_F^2 \le 4(2c_{x\epsilon}+\gamma^2)\|\mathbf{W}^k\|_F^2 + 4\gamma^2\|\mathbf{W}^{\text{init}}\|_F^2 \tag{51}$$

$$\le 8(2c_{x\epsilon}+\gamma^2)\|\mathbf{W}^k - \mathbf{W}^*\|_F^2 + 8(2c_{x\epsilon}+\gamma^2)\|\mathbf{W}^*\|_F^2 + 4\gamma^2\|\mathbf{W}^{\text{init}}\|_F^2. \tag{52}$$

Combining all of this, and taking the expectation over all steps before $k + 1$, gives us

$$\mathbb{E}\left\|\mathbf{W}^{k+1} - \mathbf{W}^*\right\|_F^2 \leq \left(1 - \gamma\eta_k + \eta_k^2 8(2c_{x\epsilon} + \gamma^2)\right)\mathbb{E}\left\|\mathbf{W}^k - \mathbf{W}^*\right\|_F^2 \tag{53}$$

$$+ \eta_k\sigma^2\|\mathbf{W}^*\|_F^2 + \eta_k^2\left(8(2c_{x\epsilon} + \gamma^2)\|\mathbf{W}^*\|_F^2 + 4\gamma^2\left\|\mathbf{W}^{\text{init}}\right\|_F^2\right). \tag{54}$$

If we choose $\eta_k$ such that $\eta_k \cdot \left(8(2c_{x\epsilon} + \gamma^2)\|\mathbf{W}^*\|_F^2 + 4\gamma^2\left\|\mathbf{W}^{\text{init}}\right\|_F^2\right) \leq \sigma^2$, we can simplify the result,

$$\mathbb{E}\left\|\mathbf{W}^{k+1} - \mathbf{W}^*\right\|_F^2 \leq \left(1 - \gamma\eta_k + \eta_k^2 8(2c_{x\epsilon} + \gamma^2)\right)\mathbb{E}\left\|\mathbf{W}^k - \mathbf{W}^*\right\|_F^2 + 2\eta_k\sigma^2\|\mathbf{W}^*\|_F^2 \tag{55}$$

$$\leq \left(\prod_{s=0}^{k}\left(1 - \gamma\eta_s + \eta_s^2 8(2c_{x\epsilon} + \gamma^2)\right)\right)\mathbb{E}\left\|\mathbf{W}^{\text{init}} - \mathbf{W}^*\right\|_F^2 \tag{56}$$

$$+ 2\sigma^2\sum_{t=0}^{k}\eta_t\left(\prod_{s=1}^{t}\left(1 - \gamma\eta_s + \eta_s^2 8(2c_{x\epsilon} + \gamma^2)\right)\right)\|\mathbf{W}^*\|_F^2. \tag{57}$$

If we choose $\eta_k = O(1/k)$, the first term will decrease as $1/k$. The second one will stay constant with time, and proportional to $\sigma^2$.

$\square$

### A.4 Applicability to vision transformers

In vision transformers (e.g. [28]), an input image is reshaped into a matrix $\mathbf{Z} \in \mathbb{R}^{N \times D}$ for $N$ non-overlapping patches of the input, each of size $D$. As the first step, $\mathbf{Z}$ is multiplied by a matrix $\mathbf{U} \in \mathbb{R}^{D \times 3D}$ as $\mathbf{Z}' = \mathbf{ZU}$. Therefore, an output neuron $z'_{ij} = \sum_k z_{ik}u_{kj}$ looks at $\mathbf{z}_i$ with the same weights as $z'_{i'j} = \sum_k z_{i'k}u_{kj}$ uses for $\mathbf{z}_{i'}$ for any $i'$.

To share weights with dynamic weight sharing, for each $k$ we need to connect all $z_{ik}$ across $i$ (input layer), and for each $j$ – all $z'_{ij}$ across $i$ (output layer). After that, weight sharing will proceed just like for locally connected networks: activate an input grid $j_1$ (one of $D$ possible ones) to create a repeating input patter, then activate a grid $j_2$ and so on.

### A.5 Details for convergence plots

Both plots in Fig. 4 show mean negative log SNR over 10 runs, 100 output neurons each. Initial weights were drawn from $\mathcal{N}(1, 1)$. At every iteration, the new input $\mathbf{x}$ was drawn from $\mathcal{N}(1, 1)$ independently for each component. Learning was performed via SGD with momentum of 0.95. The minimum SNR value was computed from Eq. (5). For our data, the SNR expression in Eq. (6) has $\left(\frac{1}{N}\sum_i (\mathbf{w}_i)_j\right)^2 \approx 1$ and $\frac{1}{N}\sum_i \left((\mathbf{w}_i)_j - \frac{1}{N}\sum_{i'}(\mathbf{w}_{i'})_j\right)^2 \approx \gamma^2/(1 + \gamma)^2$, therefore $-\log \text{SNR}_{\min} = 2\log(\gamma/(1 + \gamma))$.

For Fig. 4A, we performed 2000 iterations (with a new $\mathbf{x}$ each time). Learning rate at iteration $k$ was $\eta_k = 0.5/(1000 + k)$. For Fig. 5A, we did the same simulation but for $10^4$ iterations.

For Fig. 4B, network dynamics (Eq. (8b)) was simulated with $\tau = 30$ ms, $b = 1$ using Euler method with steps size of 1 ms. We performed $10^4$ iterations (150 ms per iteration, with a new $\mathbf{x}$ each iteration). Learning rate at iteration $k$ was $\eta_k = 0.0003/\sqrt{1 + k/2} \cdot \mathbb{I}[k \geq 50]$.

The code for both runs is provided in the supplementary material.

## B    Experimental details

Both convolutional and LC layers did not have the bias term, and were initialized according to Kaiming Normal initialization [53] with ReLU gain, meaning each weight was drawn from $\mathcal{N}(0, 2/(c_{\text{out}}k^2))$ for kernel size $k$ and $c_{\text{out}}$ output channels.

All runs were done with automatic mixed precision, meaning that inputs to each layer (but not the weights) were stored as float16, and not float32. This greatly improved performance and memory requirements of the networks.

As an aside, the weight dynamics of sleep/training phases indeed followed Fig. 3A. Fig. 6 shows $-\log \text{SNR}_w$ (defined in Eq. (6)) for weight sharing every 10 iterations on CIFAR10. For small learning rates, the weights do not diverge too much in-between sleep phases.

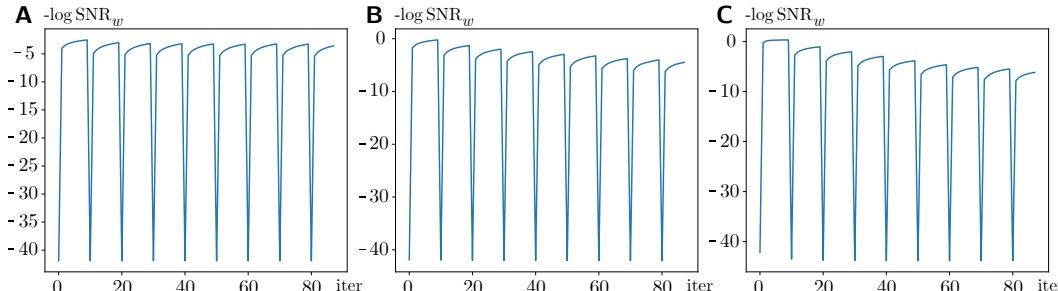

Figure 6: Logarithm of inverse signal-to-noise ratio (mean weight squared over weight variance, see Eq. (6)) for weight sharing every 10 iterations for CIFAR10. **A.** Learning rate = 5e-4. **B.** Learning rate = 5e-3. **C.** Learning rate = 5e-2.

## B.1  CIFAR10/100, TinyImageNet

Mean performance over 5 runs is summarized in Table 4 (padding of 0), Table 5 (padding of 4), and Table 6 (padding of 8). Maximum minus minimum accuracy is summarized in Table 7, Table 8, and Table 9. Hyperparameters for AdamW (learning rate and weight decay) are provided in Table 10, Table 11, and Table 12.

Hyperparameters were optimized on a train/validation split (see Section 5) over the following grids. **CIFAR10/100.** Learning rate: [1e-1, 5e-2, 1e-2, 5e-3] (conv), [1e-3, 5e-4, 1e-4, 5e-5] (LC); weight decay [1e-2, 1e-4] (both). **TinyImageNet.** Learning rate: [5e-3, 1e-3, 5e-4] (conv), [1e-3, 5e-4] (LC); weight decay [1e-2, 1e-4] (both). The learning rate range for TinyImageNet was smaller as preliminary experiments showed poor performance for slow learning rates.

For all runs, the batch size was 512. For all final runs, learning rate was divided by 4 at 100 and then at 150 epochs (out of 200). Grid search for CIFAR10/100 was done for the same 200 epochs setup. For TinyImageNet, grid search was performed over 50 epochs with learning rate decreases at 25 and 37 epochs (i.e., the same schedule but compressed) due to the larger computational cost of full runs.

## B.2  ImageNet

In addition to the main results, we also tested the variant of the locally connected network with a convolutional first layer (Table 13). It improved performance for all configurations: from about 2% for weight sharing every 1-10 iterations, to about 5% for 100 iterations and for no weight sharing. This is not surprising, as the first layer has the largest resolution (224 by 224; initially, we performed these experiment due to memory constraints). Our result suggests that adding a "good" pre-processing layer (e.g. the retina) can also improve performance of locally connected networks.

**Final hyperparameters.** Learning rate: 1e-3 (conv, LC with w.sh. (1)), 5e-4 (all other LC; all LC with 1st layer conv), weight decay: 1e-2 (all). Hyperparameters were optimized on a train/validation split (see Section 5) over the following grids. Conv: learning rate [1e-3, 5e-4], weight decay [1e-2, 1e-4, 1e-6]. LC: learning rate [1e-3, 5e-4, 1e-4, 5e-5], weight decay [1e-2]. LC (1st layer conv): learning rate [1e-3, 5e-4], weight decay [1e-2, 1e-4, 1e-6]. For LC, we only tried the large weight decay based on earlier experiment (LC (1st layer conv)). For LC (1st layer conv), we only tuned hyperparameters for LC and LC with weight sharing in each iteration, as they found the same values (weight sharing every 10/100 iterations interpolates between LC and LC with weight sharing in each iteration, and therefore is expected to behave similarly to both). In addition, for LC (1st layer conv) we only tested learning rate of 5e-4 for weight decay of 1e-2 as higher learning rates performed significantly worse for other runs (and in preliminary experiments).

For all runs, the batch size was 256. For all final runs, learning rate was divided by 4 at 100 and then at 150 epochs (out of 200). Grid search was performed over 20 epochs with learning rate decreases at

Table 4: Performance of convolutional (conv) and locally connected (LC) networks for padding of 0 in the input images (mean accuracy over 5 runs). For LC, two regularization strategies were applied: repeating the same image $n$ times with different translations ($n$ reps) or using dynamic weight sharing every $n$ batches (ws ($n$)). LC nets additionally show performance difference w.r.t. conv nets.

| Regularizer | Connectivity | CIFAR10 | | | | CIFAR100 | | | | TinyImageNet | | | |
| --- | --- | --- | --- | --- | --- | --- | --- | --- | --- | --- | --- | --- | --- |
| | | Top-1 accuracy (%) | Diff | Top-1 accuracy (%) | Diff | Top-5 accuracy (%) | Diff | Top-1 accuracy (%) | Diff | Top-5 accuracy (%) | Diff |
| - | conv | 84.1 | - | 49.5 | - | 78.2 | - | 26.0 | - | 51.2 | - |
| | LC | 67.2 | -16.8 | 34.9 | -14.6 | 62.2 | -16.0 | 12.0 | -14.1 | 30.4 | -20.7 |
| Weight Sharing | LC - ws(1) | 74.8 | -9.3 | 41.8 | -7.7 | 70.1 | -8.1 | 24.9 | -1.2 | 49.1 | -2.1 |
| | LC - ws(10) | 75.9 | -8.1 | 44.4 | -5.1 | 72.0 | -6.2 | 28.1 | 2.0 | 52.5 | 1.3 |
| | LC - ws(100) | 75.4 | -8.6 | 43.4 | -6.1 | 71.9 | -6.3 | 27.4 | 1.3 | 51.9 | 0.8 |

Table 5: Mean performance over 5 runs. Same as Table 4, but for padding of 4.

| Regularizer | Connectivity | CIFAR10 | | | | CIFAR100 | | | | TinyImageNet | | | |
| --- | --- | --- | --- | --- | --- | --- | --- | --- | --- | --- | --- | --- | --- |
| | | Top-1 accuracy (%) | Diff | Top-1 accuracy (%) | Diff | Top-5 accuracy (%) | Diff | Top-1 accuracy (%) | Diff | Top-5 accuracy (%) | Diff |
| - | conv | 88.3 | - | 59.2 | - | 84.9 | - | 38.6 | - | 65.1 | - |
| | LC | 80.9 | -7.4 | 49.8 | -9.4 | 75.5 | -9.4 | 29.6 | -9.0 | 52.7 | -12.4 |
| Data Translation | LC - 4 reps | 82.9 | -5.4 | 52.1 | -7.1 | 76.4 | -8.5 | 31.9 | -6.7 | 54.9 | -10.2 |
| | LC - 8 reps | 83.8 | -4.5 | 54.3 | -5.0 | 77.9 | -7.0 | 33.0 | -5.6 | 55.6 | -9.5 |
| | LC - 16 reps | 85.0 | -3.3 | 55.9 | -3.3 | 78.8 | -6.1 | 34.0 | -4.6 | 56.2 | -8.8 |
| Weight Sharing | LC - ws(1) | 87.4 | -0.8 | 58.7 | -0.5 | 83.4 | -1.6 | 41.6 | 3.0 | 66.1 | 1.1 |
| | LC - ws(10) | 85.1 | -3.2 | 55.7 | -3.6 | 80.9 | -4.0 | 37.4 | -1.2 | 61.8 | -3.2 |
| | LC - ws(100) | 82.0 | -6.3 | 52.8 | -6.4 | 80.1 | -4.8 | 37.1 | -1.5 | 62.8 | -2.3 |

10 and 15 epochs (i.e., the same schedule but compressed) due to the large computational cost of full runs.

Table 6: Mean performance over 5 runs. Same as Table 4, but for padding of 8.

| Regularizer | Connectivity | CIFAR10 | | | | CIFAR100 | | | | TinyImageNet | | | |
| --- | --- | --- | --- | --- | --- | --- | --- | --- | --- | --- | --- | --- | --- |
| | | Top-1 accuracy (%) | Diff | Top-1 accuracy (%) | Diff | Top-5 accuracy (%) | Diff | Top-1 accuracy (%) | Diff | Top-5 accuracy (%) | Diff |
| - | conv | 88.7 | - | 59.6 | - | 85.4 | - | 42.6 | - | 68.7 | - |
| | LC | 80.7 | -8.0 | 47.7 | -11.8 | 74.8 | -10.6 | 31.9 | -10.7 | 55.4 | -13.3 |
| Data Translation | LC - 4 reps | 82.8 | -6.0 | 50.6 | -9.0 | 76.2 | -9.2 | 35.5 | -7.1 | 58.6 | -10.1 |
| | LC - 8 reps | 83.6 | -5.1 | 53.0 | -6.6 | 77.4 | -8.0 | 35.8 | -6.7 | 59.0 | -9.7 |
| | LC - 16 reps | 85.0 | -3.8 | 55.6 | -4.0 | 78.4 | -7.0 | 37.9 | -4.7 | 60.3 | -8.4 |
| Weight Sharing | LC - ws(1) | 87.8 | -0.9 | 59.2 | -0.4 | 84.0 | -1.4 | 43.6 | 1.0 | 67.9 | -0.9 |
| | LC - ws(10) | 84.3 | -4.5 | 53.7 | -5.8 | 80.4 | -5.0 | 39.6 | -2.9 | 64.5 | -4.3 |
| | LC - ws(100) | 79.5 | -9.3 | 50.0 | -9.6 | 78.6 | -6.8 | 39.2 | -3.4 | 64.8 | -3.9 |

Table 7: Max minus min performance over 5 runs; padding of 0.

| Regularizer | Connectivity | CIFAR10 | CIFAR100 | | TinyImageNet | |
| --- | --- | --- | --- | --- | --- | --- |
| | | Top-1 accuracy (%) | Top-1 accuracy (%) | Top-5 accuracy (%) | Top-1 accuracy (%) | Top-5 accuracy (%) |
| - | conv | 0.5 | 1.0 | 1.7 | 1.0 | 0.4 |
| | LC | 0.4 | 1.6 | 1.5 | 1.0 | 1.7 |
| Weight Sharing | LC - ws(1) | 0.5 | 1.3 | 1.3 | 1.2 | 2.0 |
| | LC - ws(10) | 0.8 | 1.0 | 0.7 | 1.8 | 2.1 |
| | LC - ws(100) | 0.9 | 0.7 | 0.9 | 1.0 | 1.3 |

Table 8: Max minus min performance over 5 runs; padding of 4.

| Regularizer | Connectivity | CIFAR10 | CIFAR100 | | TinyImageNet | |
| --- | --- | --- | --- | --- | --- | --- |
| | | Top-1 accuracy (%) | Top-1 accuracy (%) | Top-5 accuracy (%) | Top-1 accuracy (%) | Top-5 accuracy (%) |
| - | conv | 0.7 | 1.5 | 0.2 | 1.2 | 1.1 |
| | LC | 0.8 | 1.1 | 0.4 | 0.7 | 0.8 |
| Data Translation | LC - 4 reps | 0.8 | 1.3 | 0.8 | 0.5 | 0.8 |
| | LC - 8 reps | 0.3 | 1.4 | 1.3 | 0.7 | 1.2 |
| | LC - 16 reps | 0.7 | 0.7 | 0.6 | 0.9 | 0.5 |
| Weight Sharing | LC - ws(1) | 0.5 | 1.1 | 0.9 | 0.9 | 0.6 |
| | LC - ws(10) | 0.6 | 1.1 | 0.3 | 0.6 | 1.2 |
| | LC - ws(100) | 0.7 | 1.0 | 0.6 | 0.2 | 0.9 |

Table 9: Max minus min performance over 5 runs; padding of 8.

| Regularizer | Connectivity | CIFAR10 | CIFAR100 | | TinyImageNet | |
| --- | --- | --- | --- | --- | --- | --- |
| | | Top-1 accuracy (%) | Top-1 accuracy (%) | Top-5 accuracy (%) | Top-1 accuracy (%) | Top-5 accuracy (%) |
| - | conv | 0.9 | 1.5 | 1.2 | 1.7 | 1.0 |
| | LC | 0.5 | 0.6 | 0.5 | 0.5 | 0.9 |
| Data Translation | LC - 4 reps | 0.4 | 0.9 | 0.3 | 0.6 | 0.8 |
| | LC - 8 reps | 0.6 | 0.9 | 0.5 | 0.5 | 0.6 |
| | LC - 16 reps | 0.9 | 0.9 | 0.6 | 0.5 | 1.1 |
| Weight Sharing | LC - ws(1) | 0.4 | 1.2 | 1.5 | 0.7 | 0.7 |
| | LC - ws(10) | 0.2 | 1.4 | 0.9 | 1.4 | 1.2 |
| | LC - ws(100) | 0.4 | 0.5 | 0.7 | 0.7 | 0.9 |

Table 10: Hyperparameters for padding of 0.

| Regularizer | Connectivity | CIFAR10 | | CIFAR100 | | TinyImageNet | |
| --- | --- | --- | --- | --- | --- | --- | --- |
| | | Learning rate | Weight decay | Learning rate | Weight decay | Learning rate | Weight decay |
| - | conv | 0.01 | 0.01 | 0.01 | 0.01 | 0.005 | 0.01 |
| | LC | 0.001 | 0.01 | 0.001 | 0.01 | 0.001 | 0.0001 |
| Weight Sharing | LC - ws(1) | 0.001 | 0.01 | 0.001 | 0.01 | 0.001 | 0.0001 |
| | LC - ws(10) | 0.0005 | 0.01 | 0.0005 | 0.0001 | 0.0005 | 0.01 |
| | LC - ws(100) | 0.0001 | 0.01 | 0.0001 | 0.01 | 0.001 | 0.0001 |

Table 11: Hyperparameters for padding of 4.

| Regularizer | Connectivity | CIFAR10 | | CIFAR100 | | TinyImageNet | |
|---|---|---|---|---|---|---|---|
| | | Learning rate | Weight decay | Learning rate | Weight decay | Learning rate | Weight decay |
| - | conv | 0.01 | 0.0001 | 0.01 | 0.01 | 0.005 | 0.0001 |
| | LC | 0.001 | 0.0001 | 0.0005 | 0.01 | 0.0005 | 0.0001 |
| Data Translation | LC - 4 reps | 0.001 | 0.01 | 0.001 | 0.01 | 0.0005 | 0.01 |
| | LC - 8 reps | 0.0005 | 0.01 | 0.0005 | 0.0001 | 0.0005 | 0.01 |
| | LC - 16 reps | 0.0005 | 0.01 | 0.0005 | 0.01 | 0.0005 | 0.01 |
| Weight Sharing | LC - ws(1) | 0.001 | 0.01 | 0.001 | 0.0001 | 0.001 | 0.01 |
| | LC - ws(10) | 0.0005 | 0.01 | 0.0005 | 0.01 | 0.001 | 0.0001 |
| | LC - ws(100) | 0.0005 | 0.01 | 0.0005 | 0.01 | 0.001 | 0.01 |

Table 12: Hyperparameters for padding of 8.

| Regularizer | Connectivity | CIFAR10 | | CIFAR100 | | TinyImageNet | |
|---|---|---|---|---|---|---|---|
| | | Learning rate | Weight decay | Learning rate | Weight decay | Learning rate | Weight decay |
| - | conv | 0.01 | 0.01 | 0.01 | 0.01 | 0.005 | 0.01 |
| | LC | 0.001 | 0.01 | 0.0005 | 0.0001 | 0.001 | 0.01 |
| Data Translation | LC - 4 reps | 0.0005 | 0.01 | 0.001 | 0.0001 | 0.0005 | 0.01 |
| | LC - 8 reps | 0.001 | 0.01 | 0.0005 | 0.0001 | 0.0005 | 0.0001 |
| | LC - 16 reps | 0.0005 | 0.0001 | 0.0005 | 0.01 | 0.0005 | 0.01 |
| Weight Sharing | LC - ws(1) | 0.001 | 0.0001 | 0.001 | 0.01 | 0.001 | 0.01 |
| | LC - ws(10) | 0.0005 | 0.01 | 0.0005 | 0.0001 | 0.001 | 0.0001 |
| | LC - ws(100) | 0.0005 | 0.01 | 0.0005 | 0.0001 | 0.001 | 0.0001 |

Table 13: Performance of convolutional (conv), locally connected (LC) and locally connected with convolutional first layer (LC + 1st layer conv) networks on ImageNet (1 run). For LC, we also used dynamic weight sharing every $n$ batches. LC nets additionally show performance difference w.r.t. the conv net.

| Model | Connectivity | Weight sharing frequency | ImageNet | | | |
|---|---|---|---|---|---|---|
| | | | Top-1 accuracy (%) | Diff | Top-5 accuracy (%) | Diff |
| 0.5x ResNet18 | conv | - | 63.5 | - | 84.7 | - |
| | LC | - | 46.7 | -16.8 | 70.0 | -14.7 |
| | LC | 1 | 61.7 | -1.8 | 83.1 | -1.6 |
| | LC | 10 | 59.3 | -4.2 | 81.1 | -3.6 |
| | LC | 100 | 54.5 | -9.0 | 77.7 | -7.0 |
| 0.5x ResNet18 (1st layer conv) | LC | - | 52.2 | -11.3 | 75.1 | -9.6 |
| | LC | 1 | 63.6 | 0.1 | 84.5 | -0.2 |
| | LC | 10 | 61.6 | -1.9 | 83.1 | -1.6 |
| | LC | 100 | 59.1 | -4.4 | 81.1 | -3.6 |