# OpenReview forum: "Towards Biologically Plausible Convolutional Networks"
_NeurIPS.cc/2021/Conference — NeurIPS 2021 Poster_

### Official Review · Reviewer_M4Vu · 2021-07-09

**Rating:** 7
**Confidence:** 3

**Summary:**

The paper presents a way to increase biological plausibility in convolutional neural networks. In particular, this paper focuses on two alternatives to weight sharing that aim at the same regularization principle. In particular, the authors propose data augmentation via multiple image translations and, much more relevant, dynamic weight sharing via lateral connections, which allows neurons to share weight information during a sleep-like phase. The paper is clear, well-structured, and I think it has value and interest for the community.

**Limitations And Societal Impact:**

Yes, they do, at the end of Section 4.

**Main Review:**

The paper presents a way to increase biological plausibility in convolutional neural networks. In particular, this paper focuses on two alternatives to weight sharing that aim at the same regularization principle. In particular, the authors propose data augmentation via multiple image translations and, much more relevant, dynamic weight sharing via lateral connections, which allows neurons to share weight information during a sleep-like phase. The paper is clear, well-structured, and I think it has value and interest for the community.

My main concern, or question/comment, is related with the main motivation of this paper: What is the problem with the fact that ConvNets are biologically implausible or that could be incompatible with current understandings in neuro-biology? I consider that all proposals to improve deep nets training or design are welcome, but I do not see the need of explicitly searching per se for biologically realistic algorithms. As Yann LeCun would say (https://cs.nyu.edu/~yann/talks/lecun-ranzato-icml2013.pdf): "Let's be inspired by nature, but not too much". It is indeed nice to be able to emulate/imitate nature, but we mainly need to understand what is actually relevant for our practical purposes: "How do we know which details are important? How do we know which details are merely the result of evolution, and the constraints of biochemistry? For airplanes, we developed aerodynamics and compressible fluid dynamics, and we figured out that feathers and wing flapping were not crucial". From this point of view, I think that any research endeavor is worth of trying, but I would like to see a better contextualization and/or justification of why we need more biologically plausible models. For instances, the authors also remark that "we trained networks using backpropagation, which is not biologically plausible". Why is that problematic? In other words, from my point of view, being inspired by nature is fine, but that inspiration should be translated into meaningful and justified design of the method in question, which not only should make sense, but also be qualitatively novel and quantitatively advantageous with regard to previous approaches.

Related to the aforementioned comment, there are some statements in the paper that remain a bit unclear to me:
- "Such models can guide medical applications such as brain machine interface and neurological rehabilitation. However, that could also lead to the design of potentially harmful adversarial attacks on the brain". How the introduced approach could be used in neurological rehabilitation? What is an adversarial attack on the brain?
- "it achieves near convolutional performance even on hard tasks, such as ImageNet classification, making it a much more likely candidate than data augmentation for the brain". What does it mean to be a "more likely candidate [...] for the brain"?

From the methodological point of view, I have several comments/questions that, in my humble opinion, if taken into account would make the paper stronger:
- In the context of ConvNets for image classification problems, locally connected networks, if I'm not mistaken, according to Figure 1, imply the use of different filters for different regions of the image. Is that correct?
- Does this dynamic weight sharing via lateral connections slow down the training procedure? By how much?
- "The lateral connections are used only for dynamic weight sharing, and not for actual computation". What does this mean exactly?
- What is the actual difference in terms of parameters to train between the ConvNets and the LC networks employed in the experiments?

**Time Spent Reviewing:**

3

---

> ### Author Response · Authors · 2021-08-10
> **Official response**
>
> Thank you for the review! To answer your main question, we need to contextualize our work a bit more (also see paragraph 2 of the Introduction). Neuroscientists have been recently treating convolutional networks as a model of the visual system [Yamins and DiCarlo, 2016, Lindsay 2020]. It’s been a very productive model, with its overall organization and learned representations being similar to the visual stream. However, the deep learning framework and conv nets in particular are “too artificial”. To build on the airplane analogy: feathers are not crucial if we want to fly, but they are crucial to understanding how birds fly. And our goal is to understand how the brain works.
>
> Our work addresses one such “too artificial” aspect, namely weight sharing, showing that it can be circumvented in a real neural circuit, which supports conv nets as a model of the visual stream. That said, our work might be beneficial to the deep learning community in the context of neuromorphic computing that shares similar limitations as real neural circuits.
>
> Below we answer more specific questions:
> 1. The “rehabilitation” and the “adversarial attack” parts were our shot at the potential long-term impacts of our work, and in general of research on brain models. What we meant is that realistic models of brain function (and the visual system in particular) can guide both medical applications and be misused, as having a detailed computational model of a system allows us to design non-hand-crafted ways to interact with it.
> 2. As for the “more likely candidate”, we argue that it’s a more suitable strategy than data augmentation for implementing well-performing locally connected networks, and one that could actually be employed by the brain.
>
> For the last list of comments, we agree that certain parts can be better explained and we will improve that in the next version:
> 1. Yes, locally connected networks use different filters at each location. We discuss that in Sec. 3.1, but we will try to improve our explanation.
> 2. Dynamic weight sharing in our implementation resulted in about 20-30% slow down compared to pure locally connected networks. That’s due to a loop over laterally connected subgrids.
> 3. We used lateral connections for weight sharing, but they were not active during training and inference (i.e., they did not influence network’s activity). As our realistic model in Sec. 4.2 implements this connectivity via an inhibitory neuron, we can think of that neuron as being silent outside of the sleep phase.
> 4. Each locally connected layer with output images of resolution $n\\times n$ is $n^2$ larger than a corresponding convolutional layer.
>
> In addition, we conducted additional experiments for ImageNet. We managed to circumvent the memory requirements of the networks (through mixed precision training), and ran a fully locally connected 0.5x width ResNet18 (without the first conv layer as in the current version). As expected, the drop in performance was rather small (about 2% Top1 for weight sharing every 1 or 10 iterations, and 4.5% for 100 iterations, and even less for Top5 performance). We will include both results in the final version.

---

### Official Review · Reviewer_t9AE · 2021-07-12

**Rating:** 6
**Confidence:** 3

**Summary:**

The paper proposes a biologically plausible substitute for convolutional networks, which uses locally connected networks with data augmentation and a weight-sharing procedure. The experiments on some benchmark datasets show that the proposed algorithm achieves good results close to convolutional networks.

**Limitations And Societal Impact:**

Yes.

**Main Review:**

Strength

The paper proposes the weight sharing procedure that is the key point of the algorithm, which enforces the weights of each neuron to be the same and close to their average. This idea is interesting and novel. It shows from simulations the weights could converge with a relatively small number of iteration, and also gives a realistic neural dynamic model to justify the weight sharing update. The experiments on CIFAR and ImageNet data show adding weight sharing can improve the LC network performance.

Weakness

1. The weight sharing update rule is given in (4), which is also Hebbian's learning rule and effectively drives all the weights to the mean. Here z_i should be taken as the input signal from previous layer x, and (z_i - 1/N\sum z_j) should be the activation of the neuron (fire rate). It is a little bit strange that in the lateral connections, the neuron j sends its input signal z_j directly to the neuron i, instead of the activations. Maybe there are some steps omitted, but I think the authors should make the dynamics clearer.
2. I'm not sure that in the real data experiments, whether the authors implement the update rule (4) or simply compute the final solution (5), or even just take the mean of the weights. Also, if using (4) or (5), does the covariance matrix C computed from current batch or all previous batch, I was not able to find these details in the paper.
3. It can be observed from table 3 in Appendix B that the results are much worse when removing the padding, and also increase the frequency of weight sharing operation is not very helpful in improving accuracy. It shows padding plays a more important role in the model and improvements from the weight sharing procedure are relatively marginal.


--- after rebuttal ---

I appreciate the author's feedback. After reading the response and other reviews, I would like to keep my original rating for this paper.

**Time Spent Reviewing:**

5

---

> ### Author Response · Authors · 2021-08-10
> **Official response**
>
> Thank you for the review! We address all of your concerns below, and are open to a discussion.
> 1. You’re correct in your understanding of how we use $z_i$, and that neurons can’t send this information directly. But we explain how to implement Eq. 4 through an inhibitory neuron in Sec. 4.2. Our implementation ends up in a circuit that obeys Dale’s law and can work with non-negative firing rates.
> 2. We implement the final solution directly (explained just before Sec. 4.1). In earlier experiments, we used Eq. 4 during the sleep phase and obtained the same results with nearly perfect weight sharing after training (the latter illustrated by Fig. 4). Due to no changes in performance and a significant computational overhead of running weight dynamics, we switched to computing the final solution. We will do a better job emphasising this in the next version.
> 3. We don’t exactly agree that padding plays a more important role. First, the results in Table 3 show TinyImageNet performance with dynamic weight sharing is very close to convolutional networks, which can be attributed to images of larger resolution compared to CIFAR10/100 (resulting in more weight sharing). This is consistent for all padding regimes (Tables 3-5). Second, ImageNet training involved random crops of smaller than original images as data augmentations (which works as non-zero padding + random crops for other datasets). As seen in Table 2 in the main text, those augmentations alone resulted in a significant loss in performance. However, it is true that data augmentations are also crucial for good results, but this is true for conv nets as well.
>
> In addition, we conducted additional experiments for ImageNet. We managed to circumvent the memory requirements of the networks (through mixed precision training), and ran a fully locally connected 0.5x width ResNet18 (without the first conv layer as in the current version). As expected, the drop in performance was rather small (about 2% Top1 for weight sharing every 1 or 10 iterations, and 4.5% for 100 iterations, and even less for Top5 performance). We will include both results in the final version.

---

### Official Review · Reviewer_uAw7 · 2021-07-15

**Rating:** 6
**Confidence:** 3

**Summary:**

This paper designed a convolutional neural network that relaxed the weight-sharing constraint by adding local lateral connections to implement dynamic weight sharing as a "sleep phase". The idea was inspired by the biological neural network, where a strict weight-sharing mechanism does not exist. They further used a Hebbian learning rule for weight update during the sleep phase. Their results suggest such a design may have comparable performance as the traditional CNN on image recognition tasks, and it is more computationally efficient than using data augmentation (e.g., translation) to force weight-sharing.

**Limitations And Societal Impact:**

N/A.

**Main Review:**

Questions and Comments:
1. Is each dynamic weight sharing update (the "sleep phase") an iterative process? Or, more specifically, is it a one-step update as described in Eq. (4), or does it converge to Eq. (5) during one intermediate "sleep phase" before starting the next "training phase"?
2. Fig. 3A illustrates the assumption that the model design will reconcile weight divergence during separate learning and weight convergence during the sleep phase. Has the author verified whether the proposed model actually exhibits this learning behavior during training?
3. The result in Table 1 suggests that more frequent weight sharing (e.g., once every batch) shows better visual task performance. Is this because "equalizing" the weights every batch may force the learning process to avoid large divergence that cannot be well-compensated by the "sleep phase"?
4. In this work, the authors remove the weight-sharing constraint in the CNN but introduce a "sleep phase" as a weight regularization process. As the original motivation is to make the model more "brain-like", what is the analogy of this "sleep phase" in the biological neural network? Is there any inspiration about how and when this dynamic weight-sharing process should happen during the learning process?

**Time Spent Reviewing:**

4

---

> ### Author Response · Authors · 2021-08-10
> **Official response**
>
> Thank you for the review! Below we address each of your questions:
> 1. The sleep phase is an iterative process, which uses Eq. 4 at each step to converge to Eq. 5. We will clarify that in the paper.
> 2. Yes, we’ve observed that weights in locally connected networks diverge during training, and our sleep phase brings them back together. We did not include the actual weight dynamics in the paper as we thought the explanation in the Fig. 3A would be enough. We will add examples of weight dynamics in the Appendix.
> 3. The sleep phase can always equalize weights if you run Eq. 4 for long enough. We think that more frequent weight sharing allows the weights to evolve in a more consistent direction, even though they’re always shared in the end, which leads to better performance. Initially, we used Eq. 4 directly in our simulations and observed that weights converge to their mean relatively quickly for any frequency of dynamic weight sharing. In the final experiment that went into the paper, we therefore equalized the weights directly to speed up simulations, which did not change the performance (we mention this just before Sec. 4.1).
> 4. The wake-sleep phases of learning have a long history in machine learning (e.g. [Hinton et.al. 1995]) and in general it is inspired by actual sleeping. In our model, sleeping fits the sleep phase naturally as the network (i.e., the visual system) stops receiving visual inputs, but maintains some internal activity. This is supported by plasticity studies during sleep (e.g. Jha et.al. 2005, Puentes-Mestril and Aton 2017). We will add this clarification to the next version of the paper.
>
> In addition, we conducted additional experiments for ImageNet. We managed to circumvent the memory requirements of the networks (through mixed precision training), and ran a fully locally connected 0.5x width ResNet18 (without the first conv layer as in the current version). As expected, the drop in performance was rather small (about 2% Top1 for weight sharing every 1 or 10 iterations, and 4.5% for 100 iterations, and even less for Top5 performance). We will include both results in the final version.

---

### Official Review · Reviewer_eCob · 2021-07-19

**Rating:** 6
**Confidence:** 5

**Summary:**

The main point of this paper is that we can rest easy about the biological plausibility of convolutional networks, because a simple hebbian mechanism - along with a “sleep phase” - provides  a method for making initially random weights match up, i.e., makes a locally connected network convolutional. The hebbian learning mechanism moves weights at every kth interval to the same value, while keeping them from moving very far from their initial value.

Through experiments on several small datasets, the authors show that this mechanism works better than simply using lots of translations of the image to make the weights similar. In addition, the performance of the hebbian-regularized model is not far from a standard convnet, and on Tiny ImageNet, it actually performs a bit better than a standard convnet.

On ImageNet, the mechanism results in performance similar to a regular convolutional network.

I have read the authors' response, and I am going to stick with my rating. If the paper does not get into NeurIPS, I encourage the authors to continue their work and submit it somewhere else, e.g., WACV or Vision Research.

**Limitations And Societal Impact:**

The authors are clear about the limitations; the model works best when the sleep phase is implausibly frequent. The model requires fairly precise connectivity, and there is a somewhat implausible sequential application of the hebbian part, where layer l is regularized before layer l+1. While the authors suggest that this could be done simultaneously, but would take longer, this is not demonstrated.

In terms of societal impact, I don’t see any way that this would have societal impact, and neither do the authors.


**Main Review:**

Originality: As far as I know, this is original.

Quality: The work appears sound. There is a proof of the dynamics of the hebbian part in the appendix that I didn’t check. The claims in the abstract are well supported by the results.

 Clarity: Is the submission clearly written? Yes: the paper is quite clear.

Significance: Are the results important? To the extent that they give a biologically-plausible mechanism for generating nearly-convolutional networks, yes. One has to care about such things; I do. The work provides a justification of layered convolutional networks as models of the visual system.

Minor comments:
line 227: it is to frequent -> it is too frequent.

**Time Spent Reviewing:**

1.5

---

> ### Author Response · Authors · 2021-08-10
> **Official response**
>
> Thank you for the review! We’ve fixed the minor comment, and would like to elaborate on the limitations mentioned in your review.
> 1. The model indeed works best with very frequent sleep phases, but the performance drop for less frequent ones is still small. We would also like to point out that slower learning rates should enable less frequent sleep phases (as the weights would diverge less). While it is a bad strategy in deep learning due to increased training times (e.g. making it 10x slower would require well over a month of training on ImageNet in our setting), we imagine it is consistent with animal learning (especially if we depart from supervised learning).
> 2. We discuss the need for precise lateral connectivity in Sec. 6; it would be interesting to propose a more detailed model for learning lateral connectivity, but that would deserve its own paper.
> 3. We omit experiments on sequential sleep phases due to the properties of weight dynamics in Eq. 4: as long as $\\mathbb{w}^{\\mathrm{init}}$ remains the same, any changes in $\\mathbb{w}$ will not affect the final solution -- as long as the layer ends up receiving repeating inputs $\\mathbb{x}$. Therefore, the deeper layers would converge to the correct solution once the shallower layers have done so.
>
> We will elaborate on these limitations in the main text (especially the last one). Could you direct us to any additional concerns that you have that would prevent you from raising your score?
>
> In addition, we conducted additional experiments for ImageNet. We managed to circumvent the memory requirements of the networks (through mixed precision training), and ran a fully locally connected 0.5x width ResNet18 (without the first conv layer as in the current version). As expected, the drop in performance was rather small (about 2% Top1 for weight sharing every 1 or 10 iterations, and 4.5% for 100 iterations, and even less for Top5 performance). We will include both results in the final version.

---

### Author Response · Authors · 2021-08-30
**Thanks for the reviews**

We would like to thank the reviewers again for their comments! We believe that we’ve addressed all of their concerns in our responses.

---

### Decision · Program_Chairs · 2021-09-27

**Decision:**

Accept (Poster)

**Comment:**

This paper received 3 marginal accepts and 1 accept. The reviewers stated that their lack of enthusiasm had to do with the somewhat niche aspect of the work (not because of technical or experimental limitations). Many NeurIPS attendees do care about biology and the work should be of interest to this community as noted by one of the reviewers. The AC thus recommends acceptance.